# Evaluation of Novel B1R/B2R Agonists Containing TRIOZAN™ Nanoparticles for Targeted Brain Delivery of Antibodies in a Mouse Model of Alzheimer Disease

**DOI:** 10.3390/molecules28135206

**Published:** 2023-07-04

**Authors:** Maxime Gagnon, Martin Savard, Thi Minh Hue Tran, Laurence Vincent, Alexandre Moquin, Philippe Tremblay, Xavier Roucou, Yves Dory, Fernand Gobeil

**Affiliations:** 1Department of Pharmacology & Physiology, Université de Sherbrooke, Sherbrooke, QC J1H 5N4, Canada; maxime.gagnon2@usherbrooke.ca (M.G.); martin.savard@usherbrooke.ca (M.S.); laurence.vincent@usherbrooke.ca (L.V.); 2Department of Chemistry, Université de Sherbrooke, Sherbrooke, QC J1R 2R1, Canada; thi-minh-hue.tran@usherbrooke.ca (T.M.H.T.); yves.dory@usherbrooke.ca (Y.D.); 3Ovensa Innovations Inc., 101 Boulevard Cartier Ouest, Laval, QC H7Y 5B7, Canada; alexandre.moquin@hotmail.com (A.M.); ewing_21@hotmail.com (P.T.); 4Department of Biochemistry & Functional Genomics, Université de Sherbrooke, Sherbrooke, QC J1H 5N4, Canada; xavier.roucou@usherbrooke.ca

**Keywords:** nanoformulations, kinin agonist analogues, B1 and B2 receptors, biologics, blood–brain barrier, Alzheimer disease

## Abstract

The blood–brain barrier (BBB) is a major obstacle to the development of effective therapeutics for central nervous system (CNS) disorders, including Alzheimer’s disease (AD). This has been particularly true in the case of monoclonal antibody (mAbs) therapeutic candidates, due to their large size. To tackle this issue, we developed new nanoformulations, comprising bio-based Triozan polymers along with kinin B1 and B2 receptor (B1R and B2R) peptide agonist analogues, as potent BBB-permeabilizers to enhance brain delivery of a new anti-C1q mAb for AD (ANX005). The prepared B1R/B2R-TRIOZAN™ nanoparticles (NPs) displayed aqueous solubility, B1R/B2R binding capacity and uniform sizes (~130–165 nm). The relative biodistribution profiles of the mAb loaded into these NPs versus the naked mAb were assessed in vivo through two routes of administrations (intravenous (IV), intranasal (IN)) in the Tg-SwDI mouse model of AD. At 24 h post-administration, brain levels of the encapsulated mAb were significantly increased (up to 12-fold (IV) and 5-fold (IN), respectively) compared with free mAb in AD brain affected regions, entorhinal cortex and hippocampus of aged mice. Liver uptakes remained relatively low with similar values for the nanoformulations and free mAb. Our findings demonstrate the potential of B1R/B2R-TRIOZAN™ NPs for the targeted delivery of new CNS drugs, which could maximize their therapeutic effectiveness.

## 1. Introduction

Alzheimer’s disease (AD) is a progressive neurodegenerative disorder affecting elderly people worldwide and is mainly characterized by two major pathological hallmarks in the AD brain: extracellular plaques, containing various forms of amyloid-β protein (Aβ), and intracellular neurofibrillary tangles (NFTs), composed of hyper-phosphorylated tau protein [1]. However, recent failures in drug development, along with fact that the only available symptomatic drugs currently on the market for AD have limited effects on disease progression, leave a large unmet medical need [2]. As such, there is a consensus among the community for the need to discover new, more efficient therapeutic targets in a multi-combination approach that ultimately aims to stop or reverse early stages of disease progression.

AD immunotherapeutic drugs, such as monoclonal antibodies (mAbs), have been developed rapidly in the last decade. The major challenges of these immunotherapeutic approaches are poor blood–brain barrier (BBB) penetration, induction of inflammation and microhemorrhages, absence of significant cognitive effects, and off-target reactivity [2,3,4,5,6,7]. One key area of current investigations is related to the involvement of the immune system in AD. More specifically, various preliminary data have shown the important contribution of both microglia and astrocytes to AD-related synapse loss (with neuronal damage and neuroinflammation) where a key role is played by the complement pathway, specifically by the C1q initiator protein in parallel with oligomeric Aβ, as the mechanism by which glial cells modulate synapse pruning [8,9]. In that perspective, targeting C1q as the initiator of the classical complement pathway to block its activation (including downstream complement C3 and C5), while leaving the protective functions of other complement pathways (e.g., lectin and alternative pathways) intact, may represent a potential breakthrough therapeutic intervention in AD [5,7]. One good example is the successful development of the C1q function blocking humanized IgG4 mAb, ANX005 (Annexon Biosciences), which has limited ability to penetrate the BBB on its own but has shown promising therapeutic effects in preclinical models of autoimmune and neurodegenerative diseases, including AD [10,11,12,13,14]. Taken together, there is thus a major need to optimize future AD immunotherapeutics, including those targeting the classical complement pathway, so as to provide improved brain delivery capabilities with enhanced efficacy and safety profiles for the prevention or delay of AD.

An emerging and promising field for drug delivery across the BBB involves nanocarrier-based systems [11,12,13,14]. Among all drug delivery nanosystems currently undergoing active research and approved for biomedical applications, there are chitosan- and chitosan derivative-based nanocarriers. Prominent among the latter is TRIOZAN™, which uses a single hydrophilic quaternized biopolymer *^N,N,N^*-trimethyl chitosan (TMC) [11,12,13,14]. The bio-based, self-assembled TMC nanocarriers allow efficient encapsulation and protection of drug molecules against degradation while simultaneously maintaining therapeutic integrity. TMC has been shown to be readily soluble in water (over a wide range of pH), safe, biodegradable, biocompatible, and can be easily chemically conjugated for active targeting of the loaded therapeutic agent [11,12,14]. Moreover, because of its good muco-adhesive properties, it has been used to develop nanocarriers for brain targeting via intranasal administration, in addition to standard intravenous injections [15,16]. 

Yet another strategy to enhance drug delivery to the brain involves the targeted permeabilization of the BBB at inflammatory sites through the activation of overexpressed G protein-coupled receptors (GPCRs), which includes the kinin B2 and B1 receptors (B2R and B1R). Indeed, metabolically resistant B1R or B2R selective kinin peptide agonists have been shown to be effective in improving the delivery of drugs into the CNS via transient opening of the BBB with no significant side effects, such as brain edema [17,18,19,20,21]. Moreover, our previous work has demonstrated that the combined use of biostable B1R and B2R agonists has additive effects on the modulation of BBB paracellular permeability to enhance drug delivery at diseased sites, therefore enhancing both their specificity and their therapeutic effects [19,21,22,23]. Interestingly, some studies have reported higher expression of B1R and B2R in the cerebral microvasculature of AD transgenic (Tg-SwDI) mice compared with age-matched normal mice [24,25]. This observation could be attributable to the production and action of pro-inflammatory cytokines in areas involved in AD, in particular, IL-1β and TNF-α, both recognized as potent inducers of B1R and B2R expression [25,26,27]. 

Herein, we develop and evaluate a new type of nanodelivery system based on kinin-B1R/B2R agonist-coated NPs to promote effective transvascular drug delivery across the pathological BBB, which could lead to increased brain penetration and therapeutic potency of the mAb candidate ANX005. The relative biodistribution profiles of the mAb loaded into these NPs versus the naked mAb were assessed in vivo through two routes of administrations (intravenous (IV), intranasal (IN)) in the Tg-SwDI mouse model of AD. 

## 2. Results and Discussion

The ANX005 shows promise as a potential treatment for several neurodegenerative disorders, such as age-related macular degeneration (AMD) and AD [8,28]. In fact, there are several completed and ongoing clinical studies evaluating its effectiveness in mitigating/preventing the development of neurodegenerative diseases (e.g., Guillain-Barré syndrome (https://clinicaltrials.gov Identifier: NCT04035135, accessed on 2 July 2023), Huntington’s Disease (https://clinicaltrials.gov Identifier: NCT04514367, accessed on 2 July 2023), amyotrophic lateral sclerosis (ALS) (https://clinicaltrials.gov Identifier: NCT04569435, accessed on 2 July 2023)). In the present study, we used the murine version of ANX-005, referred to as ANX-M1 (Annexon Biosciences). The latter was generated by immunizing C1qa KO mice with human C1q protein purified from human plasma using standard mouse immunization and hybridoma screening methods, as described by Hong et al. (2016) [8]. The schematic representation and amino acid sequence of the mAbs ANX-M1 can be found in Appendix A.

Chemical synthesis of kinin agonist-functionalized TRIOZAN™ NPs was accomplished in the following way (Figure 1). First, the agonist sequence was prepared on solid support using standard SPPS procedures (Fmoc strategy, step 1). In the same way the two-residues Mpa-βAla spacer was introduced (step 2). In a third step the full spacer–agonist peptide was released from the resin as its free thiol. Finally (step 4), the free thiol was coupled at the 6-OH position of the TRIOZAN™ subunits that had been freshly activated as its corresponding tosylates (S_N_2 reaction leading to thioethers), thus yielding the desired TRIOZAN™ polymers. According to the number of equivalents of peptide-free thiol used during step 4, the degree of conjugation varies. 

Our results show that the conjugation of TRIOZAN™ to the N-termini of biostable peptide B1R agonist (NG29) and antagonist (R954) covalently extended with Mpa/βAla spacer may be tolerated, providing TMC–peptide conjugates with a ratio of 1:7 with high affinities towards the human B1R (hB1R) (see IC_50_ values in the low nanomolar range of compounds TRIOZAN™-MAB7108 and TRIOZAN™-MAB7122, Table 1). By comparison, TMC–B1R agonist conjugates prepared with a 1:1 stoichiometry led to a ~400-fold reduction of the binding affinity toward hB1R (IC_50_ value: 200 nM) compared with the parent unmodified peptide agonist, hence, this option was discarded. When applied to the biostable B2R agonist NG291, the N-terminal extension with Mpa was found to have detrimental effects on its binding capacity to hB2R (MAB7053 (IC_50_ value: 1180 nM) vs. NG291 (IC_50_ value: 2 nM); Table 1). Such negative effects were also observed with the use of other types of linkers (e.g., PEG2, PEG8, Gly) (not shown). Moreover, the bioconjugation procedure of TMC and N-terminal Mpa-NG291, with the optimal ratio found to be 1:7 for the B1R agonist coupling to TMC, led to a further decrease of the resulting TMC–B2R conjugate for its B2R target (TRIOZAN™-MAB7053 IC_50_ value: 10,600 nM, Table 1). To circumvent the possible steric hindrance effect of the linker affecting NG291–B2R interaction, we adopted the strategy of creating B1R-functionalized TRIOZAN™ NPs complexed with free (non-cross-linked) B2R agonists for delivering the mAbs across the BBB (nanoformulation F4). Another version of dual B1R/B2R peptide NPs comprising a mix of unconjugated B1R and B2R agonists was prepared to potentially maximize the ligand biological activity of both (nanoformulation F6) (see Figure 1 and Figure 2 for schematic representations of the two types of NPs).

TRIOZAN™ is a positively charged polymer owing to its numerous quaternary ammonium groups. It can be electrostatically cross-linked by adding negatively charged polytriphosphate (3PP) leading to the formation of stable NPs that act as nanocontainers. Their main payload is the mAb, which is non-covalently retained inside the NPs until they reach the desired location for delivery. This selectivity is provided by B1R and B2R ligands. Similar to the mAb, these ligands can be non-covalently trapped inside the NPs, from which they can slowly leach out to bind to the cognate receptors. This corresponds to nanoformulation F6, for which the NPs contain twice as much B1R ligand as B2R ligand. This was done to limit the potential hypotensive effects of NPs containing NG291 agonists, mostly attributable to systemic vascular B2R activation [19,21,23]. Alternatively, the ligands can be covalently attached to the NPs by a spacer, meaning that binding between receptors and ligands can be somehow hindered by the bulk of the NPs. 

Binding experiments with HEK-293 cells stably expressing human B1R (Figure 1A) or B2R (Figure 1B) have also been performed with new unloaded, peptide B1R agonist-functionalized TRIOZAN™ NPs mixed with B2R agonists. Results showed that the NPs, prepared using the specified starting material described in Table 1 (referred to below as F4 nanoformulation, see Table 2), retained their abilities to bind to human B1R and B2R and to fully displace selective radioligands. 

The mAb liquid solution (F1; 2.5 mg/mL) and the new TRIOZAN™ NP formulations loaded with mAb, F4 and F6, in lyophilized forms (in PBS/sucrose) were readily dispersible in deionized water. The chosen 2.5 mg/mL of F1 corresponds to a dose of 10 mg/kg by injecting a volume of 4 mL/kg (ex. 80 μL for a 20 g mouse). A summary analysis of the characteristics of the mAb solution and the water-soluble NP formulations is presented in Table 2 (see also Appendix A for representative DLS graphs). The final concentrations of loaded mAb in F4 and F6 NPs were in the same range as the original mAb solution (2.5–3.3 mg/mL). The mean diameters of NPs were found to be less than 200 nm with polydispersity index (PDI) values below 0.3, indicating monodisperse NP suspensions. Due to their favorable characteristics (aqueous solubility, B1R/B2R binding capacity, uniform and small sizes (i.e., between 130 and 165 nm)), the new NPs were considered appropriate for in vivo studies.

Prior to performing the mAb tissue biodistribution experiments, an acute, single-dose toxicity assessment of unloaded B1R/B2R-targeted NPs (F4 and F6) was carried out in conscious adult normal mice. No sign of abnormal behavior or distress was observed upon their intravenous (IV) tail vein injection during a period of 24 h. 

Biodistribution studies were then performed to characterize the relative effects of the two NP preparations on the accumulation of the mAb ANX-M1 at specific regions of the brain, such as the entorhinal cortex and the hippocampus (regions responsible for learning and memory), most likely to be affected in AD. Given the relatively long half-life (t½: ~18 h) in rodents for ANX-005 [9], the comparison of the biodistribution profiles of the free mAb and NPs was evaluated at 24 h post-injection. For the purpose of these experiments, we developed a new, highly sensitive LC-MS/MS assay for a direct quantification of the mAb ANX-M1 in blood and tissue extracts (specifically, brain, liver, serum) from transgenic AD adult mice (see Materials and Methods). The free mAb (F1) and the new NP formulations containing the mAb encapsulated with the B2R agonist in B1R agonist-conjugated TRIOZAN™ NPs (F4), or the mAb encapsulated with the B1R and B2R agonists inside TRIOZAN™ NPs (F6), were administered by IV injection and organ-to-serum ratios of mAb were quantified (Figure 2). IV injections of F4 and F6 led to ~4- and 12-fold increases, respectively, in the brain delivery of mAb compared with F1 in AD mice, indicating the cerebral bioavailability enhancement offered by these two formulations. Furthermore, there was no apparent sex difference in the brain uptake of the tested formulations given IV (not shown). Looking specifically at F6, the amounts of mAb found in the brains were about 60-fold higher than that in livers. As seen in Figure 2, liver uptakes remained relatively low, with similar values for the two tested formulations. Interestingly, despite the final concentration of the B1R agonist being the same between the F4 and F6 (i.e., 90 µM), the latter resulted in significantly more mAb accumulation in both the hippocampus and the entorhinal cortex regions (Figure 2). This suggests that the complexation rather than the covalent coupling of B1R agonists to TRIOZAN™ is more suited to increasing the permeability of the BBB by the B1R. Such inference is supported by the binding data showing a reduced binding affinity of the B1R agonist (~15-fold lower) when tethered onto TRIOZAN™ as compared with the unmodified agonist (see Table 1). 

In any case, it is likely that the higher, specific brain accumulations of the mAb that were achieved with the different nanoformulations that resulted from the enhanced cerebrovascular permeability mediated by the B1R/B2R agonist analogues assembled with TRIOZAN^TM^-based NPs [17,20,21]. 

The intranasal (IN) administration route has recently gained increasing interest for delivering drugs to the CNS while bypassing the BBB [16,29,30]. Therefore, for comparison purposes, we investigated the potential of the new drug-loaded nanoformulations to distribute into the brain after IN administration. For this, the naked mAb alone (F1) and the mAb-loaded NP formulations (F4 and F6) were administered by intranasal drops at the same mAb dosage of 10 mg/kg. The nanoformulations were well tolerated with no apparent adverse effects for up to 24 h after their administration (not shown). At 24 h post-delivery, significant increases in the mAb levels were observed in the entorhinal cortices and hippocampi of mice treated with the F4 (3- to 4-fold) and F6 (4-to 5-fold) versus F1 (Figure 3). Systemic exposure was low and highly variable for all formulations upon their IN administrations (not shown). 

Finally, a comparison of the brain biodistribution of the mAb contained in F1, F4, and F6, following IV and IN administrations in AD transgenic mice revealed that the systemic IV mode of administration is highly superior to IN for the targeted delivery of both free mAbs and the mAbs-loaded TRIOZAN™ NPs at AD brain regions (ranging from ~5- to 15-fold higher) (Figure 4). 

Altogether, our findings demonstrate the efficacy of dual B1R/B2R agonists/TRIOZAN™-based NPs to enhance the targeting of mAbs to the brain. Moreover, our results are consistent with other studies that use different types of kinin agonist-functionalized NP-based therapies selectively targeting either the B1R or the B2R in CNS-related diseases, including glioblastoma [31], viral encephalitis [32] and AD [33]. Further studies are warranted to determine whether the desirable pharmacokinetic property of the new nanodelivery system translates to the improved pharmacodynamic potential of the new anti-C1q mAb as well as to other new biologics candidates.

## 3. Conclusions

Almost all antibody-based immunotherapies have poor brain bioavailability, which limits their use for the treatment of CNS diseases. In this study, we engineered novel TRIOZAN™-based NPs comprising potent B1R and B2R agonists to deliver ANX-M1, a new anti-C1q mAb candidate for the treatment of AD, to the brain. Our results indicate that while a certain quantity of the mAb was able to permeate the perturbed BBB on its own in AD mice, its delivery across the BBB and to brain regions related to cognitive control can be significantly enhanced via the use of the newly developed nanodelivery system. In fact, the delivery of ANX-M1 in different areas of the brain was significantly increased (up to 12 times) when it was encapsulated in the new B1R/B2R targeting NPs (F6), when compared with its administration as a naked mAb (F1). The systemic IV delivery of the new nanoformulations appeared safe, with no apparent sign of toxicity, and is recommended over the IN administration for achieving the highest level of delivery of immunotherapy candidates to the diseased brain. The dual B1R/B2R agonists containing NPs with BBB-modulating capabilities could provide a new avenue for increasing the brain distribution and, possibly, the therapeutic efficacy of diverse biologics.

## 4. Materials and Methods

### 4.1. Peptide Synthesis and Purification

Peptides were assembled on 2-chlorotrityl-chloride resin (0.9 mmol/g) by an automated Symphony-X peptide synthesizer (Protein Technologies, Inc., Tucson, AZ, USA) using standard 9-fluorenylmethyloxycarbonyl (Fmoc) SPPS chemistry [21]. Briefly, the couplings of 5 equivalents of Fmoc-protected amino acids were performed in the solution of dichloromethane (DCM)/N, N-dimethylformamide (DMF) (1:1, *v*/*v*) using 5 equivalents of HATU and 9 equivalents of DIPEA for 1 h. When applicable, the N-terminal Mpa residue was introduced as its Trt S-protected derivative. Fmoc groups were removed by treating the resin with 20% piperidine in DMF for 10 min (2×). The peptides were cleaved from resin by HFIP/DCM (7/3) for 1 h and deprotected using the cleavage cocktail of TFA: triisopropylsilane (TIPS): H_2_O (95:2.5:2.5/*v*:*v*:*v*) at room temperature for 3 h. The solution of the released peptides was filtered, concentrated, and precipitated in cold diethyl ether. After centrifugation, the supernatants were removed and thereafter, crude peptides were purified by analytical RP-HPLC (Waters 2535 module) on a C18 column (ACME C18, 10 µm, 250 × 30 mm, Canadian Life Science, Peterborough, ON, Canada) using absorbance at 214 nm. Purified peptide fractions were pooled, lyophilized, and stored at −20 °C. Stock solutions (10 mM) of peptides were also prepared in Nanopure water and then stored at −20 °C until use. The purity and identity of purified peptides were evaluated using ultra performance liquid chromatography–tandem mass spectrometry (UPLC-UV-MS, Waters AQUITY-H-Class-SQD2, column Waters BEH C18 (1.7 µm, 2.1 × 50 mm)). All synthesized peptides were >98% pure, with expected mass spectra. Abbreviations for amino acids follow the recommendations of the IUPAC-IUB Commission on Biochemical Nomenclature. Other abbreviations are described as follows: Hyp, trans-4-hydroxy-L-proline; Thi, α-(2-thienyl)-L-alanine; Orn, L-ornithine; Oic, L-(2S,3aS,7aS)-octahydro-1H-indol-2-carboxylic acid; β-Nal, 3-(2-naphthyl)-alanine; Igl, 2-indanyl-glycine; ^N^Chg, N-cyclohexyl-glycine; Sar, N-methyl-glycine; (αMe)Phe, α-methyl-phenylalanine; Mpa, 3-mercaptopropanoyl.

### 4.2. Peptide Conjugation on TRIOZAN™ 

One hundred mg of Ts-TRIOZAN™ (TMC: *^N,N,N^*-trimethyl chitosan, M.w. = 65,000 Da, degree of quaternization: 60%, provided by Ovensa Innovations Inc., Laval, QC, Canada), 1 mL of TEA and 13 mg B1R agonist peptide were dissolved into 15 mL of water in a 100 mL round-bottom flask under inert condition (7 equivalent peptide/Triozan; TMC-B1R). The reaction mixture was kept under stirring at room temperature for 3 days. After the grafting reaction, the mixed solution was dialyzed against deionized water using 12–14 kDa cut-off dialysis membranes (Spectrum LabsTM) for 3 days at room temperature (with bath changes every day), then recovered by lyophilization and stored at −20 °C until further utilized.

### 4.3. Preparation of TRIOZAN™-Based NPs for mAb Loading

TRIOZAN™ chloride powder (TMC) was dissolved in autoclaved acetate buffer saline (pH 5.6, 10 mM, NaCl = 137 mM) to a final concentration of 3 mg/mL, and then filtered using a media filtration system (pore size: 0.45 μm). TRIOZAN™ solution was adjusted to pH > 9 using autoclaved bicarbonate buffer saline solution (pH 9.3, 0.1 M, NaCl 137 mM) by adding 20% volume. The mAb antibody stock solution (2.5 mg/mL in phosphate-buffered saline (PBS)) was diluted 1.2× with bicarbonate buffer saline (pH 9.3, 0.1 M, NaCl 137 mM) and incubated 10 min on ice. Thereafter, mAb was added dropwise into TRIOZAN™ to a final weight ratio of 1:30 antibody: TRIOZAN™. The peptide B1R agonist (NG29, final concentration 90 μM) and the B2R agonist (NG291, final concentration 45 μM) were each subsequently added for Formulation 6. For the preparation of the Formulation 4, comprising B1R agonist-functionalized NPs, a TMC/TMC-B1R ratio of 9/1 was used instead of the free B1R peptide agonist. The TRIOZAN™ (+mAbs/B1R/B2R) solution and a solution of PBS (pH 7.4, 0.01 M, NaCl 137 mM) containing dextran sulfate (DS, M.w. = 500,000 Da: 1 mg/mL): tripolyphosphate (TPP: 0.5 mg/mL), were loaded in two separate 10 mL syringes. Both syringes were placed in a NanoAssemblr containing a sterile NanoAssemblr NxGen microfluidic chip. The flow rate was set at 12 mL/min with a final wt/wt ratio between TRIOZAN™ and DS equal to 6:1. The freshly prepared NPs were centrifuged at 42,000× *g* for 45 min at 4 °C. The pellets (clear with a yellowish tint) were resuspended in PBS using short bursts of sonication and filtered through 0.45 μm PVDF filters. Sucrose (final concentration: 50 mg/mL) was finally added to NP suspensions before freeze-drying.

### 4.4. Competition Binding Assays

Competition binding assays were carried out essentially as described previously [20]. Briefly, HEK293 cells stably expressing either human kinin B1R or B2R were grown in 24-well plates and incubated with 4 nM [^3^H]Lys[Leu^8^]desArg^9^BK (for B1R) or with 1 nM [^3^H]BK/well (for B2R) in serum-free DMEM for 1–2 h in the absence or presence of increasing concentrations of competitors (10^11^–10^−5^ M for peptides; 10^−10^–10^−4^ g/mL for peptides-coupled NPs). Radioactivity of samples was measured by a β-scintillation counter (PerkinElmer, Woodbridge, ON, Canada). Binding affinities of compounds were expressed in terms of IC_50_ values—the concentrations of competing compounds (i.e., free peptides, TMC-peptides conjugates and peptides coupled to NPs) causing 50% displacement of specific binding of the radioligand. Data are means of 2–3 independent experiments.

### 4.5. mAb Protein Determination

Absolute protein concentrations of mAbs alone or loaded in TRIOZAN™-based NP formulations were determined using the NanoDrop ND-1000 spectrophotometer (Thermo Fisher, Waltham, MA, USA). Samples were measured for light absorbance at 280 nm and Beer’s Law was used to calculate concentration, with molar extinction coefficient (ε) equal to 210,000 M^−1^ cm^−1^.

### 4.6. DLS Analysis

The average sizes (nm) and polydispersity indices of TRIOZAN™-based NPs were determined by dynamic light scattering (DLS) using the UNCLE-Unchained Labs system (Unchained Labs, Pleasanton, CA, USA). Ten µL of each antibody was loaded into the multi-micro cell array. The DLS measurement was recorded at 20 °C. Ten readings were taken for each individual analysis, with outliers discarded, and the remaining data averaged. The UNCLE software (version 2.0) correlation function was subsequently used to calculate DLS measurements (size distribution and polydispersity). Alternatively, DLS analyses of the mAbs-loaded TRIOZAN™ NPs were performed at room temperature using a Zetasizer Nano ZS (Malvern^®^ Instrument, Malvern, United Kingdom).

### 4.7. Animals

Male and female double transgenic mice (B6C3-Tg (APPswe/PSEN1dE9); 9- to 12-month-old; bodyweight 35–50 g) were used in this study. Mice were purchased from the Jackson Laboratory (MMRRC Strain #034829-JAX). Animals were maintained under standard diurnal conditions and were allowed access to food and water ad libitum. Animal experiments were approved by the Institutional Animal Care and Use Committee of the Université de Sherbrooke (protocol #2020-2508) and performed in accordance with the Canadian Council on Animal Care guidelines.

### 4.8. Biodistribution Studies

Biodistribution studies of the naked mAb and the two ANX-M1 mAb-loaded NP formulations were conducted following two routes of administration (IV, IN) in the transgenic mouse model of AD. Tissue samples (entorhinal cortex, hippocampus, and liver) and blood were collected 24 h post-administration. The ANX-M1 mAbs was detected by targeted proteomics on its unique peptide sequence: SSGYHFTSYWMHWVK (see Appendix A). a stable heavy isotope-labeled counterpart (AQUA peptide: SSGYHFTSYWMHWV[K + 8]) was used as an internal standard (ThermoScientific). A standard curve was established with a synthetic peptide and the heavy labelled peptide in a peptide mixture at various concentrations. Absolute quantification experiments were conducted in tissue samples and the serum (see text below).

### 4.9. mAb Administration and Tissue Collection

mAb formulations (dose: 10 mg/kg ANX-M1) were administered IV or IN in isoflurane-anesthetized mice (5% induction, 2% maintenance). For IV administration, formulations were given as a bolus (<1 min) with a 50 μL saline flush via the caudal vein. For IN administered formulations, after rehydration of lyophilized powders, NP formulations were concentrated about 6- to 7-fold using 10 kDa MWCO ultrafiltration membranes (Amicon Ultra, Oakville, ON, Canada). A total volume of 5–6 μL was delivered into each nostril using a pipette over the course of 5 min. At the end of the experiments, blood and organs were collected from mice anesthetized with ketamine/xylazine (intramuscular injection, 87/13 mg/kg). Serum was prepared from blood harvested from cardiac puncture. Prior to organ harvesting, mice were euthanized by transection of the right atrium followed by an injection of 20 mL cold saline in the left ventricle to flush the blood from the brain and organs. Harvested tissue and serum samples were flash frozen in liquid nitrogen and stored at −80 °C until analysis. 

### 4.10. Mass Spectrometry Sample Preparation

Tissue samples were prepared according to Delcourt et al. (2018) and Dubois et al. (2020) [34,35]. Briefly, tissue samples were homogenized using a Mini Bead Mill Homogenizer with 2.8 mm ceramic beads for hard tissue homogenization in 500 μL lysis buffer (8 M urea, 10 mM HEPES pH 8.0). Lysates were sonicated to reduce viscosity followed by a 10 min centrifugation at 16,000× *g* to discard debris and insoluble parts. The supernatant was transferred in a LoBind tube and protein content was assessed using BCA protein assay. A total of 50 μg of protein in 50 μL of lysis buffer with 20 nM of the stable isotope peptide were reduced in 5 mM DTT, boiled 2 min at 95 °C, rested 30 min at room temperature and alkylated in 7.5 mM 2 Choloroacetamide for 30 min in the dark at room temperature. The urea concentration in the lysate was reduced to 2 M with the addition of 50 mM NH_4_HCO_3_ and the samples were subjected to overnight trypsin digest (Trypsin Gold, MS Grade, Promega Corporation, Madison, WI, USA). Following digestion, the extracted peptides were acidified to a final concentration of 0.2% trifluoroacetic acid (TFA), desalted using C18 Zip-Tips, eluted with 200 μL of acetonitrile (ACN): H_2_O: formic acid (FA) (50:49:1), dried in a speedvac, and resuspended in 100 μL 1% FA in water. 

#### 4.10.1. LC-MS/MS Analysis

A total of 10 μL of peptide mixture was loaded and separated onto a nanoHPLC system (Dionex Ultimate 3000) with a constant flow of 4 μL/min onto a trap column (Acclaim PepMap100 C18 column, 0.3 mm id × 5 mm, Dionex Corporation, Sunnyvale, CA, USA). Peptides were then eluted off towards an analytical column heated to 40 °C (PepMap C18 nano column (75 μm × 25 cm)) with a linear gradient of 5–45% of solvent B (80% ACN with 0.1% FA) over a 42 min gradient at a constant flow (450 nL/min). Peptides were analyzed on an OrbiTrap QExactive (Thermo Fischer Scientific) spectrometer using an EasySpray source at a voltage of 2.0 kV using PRM method. Acquisition of the MS/MS spectra was performed in the Orbitrap. An inclusion list containing the m/z values corresponding to the monoisotopic form of the normal and equivalent AQUA peptide (639.2910/641.9624 (3+) and 479.7201/481.7236 (4+)) of the mAb was generated. The collision energy was set at 28% and resolution for the MS/MS was set at 35,000 for 200,000 ions with a maximum filling time of 110 ms with an insulation width of 2 and a chromatography peak width of 30s. Data acquisition was undertaken using Xcalibur version 3.1.66.10.

#### 4.10.2. Data and Statistical Analysis

Identification and quantification of the test mAb-derived peptide were performed on Skyline software (21.2.0.425). For quantification, the 6–7 most intense fragment ions were used for either the light peptide or the heavy peptide (Table 3). Control experiments using similar brain tissues/serum samples from a mAb-untreated B6C3-Tg AD mouse revealed minimal interference from background ions (not shown). These noise values were subtracted from signal values in the MS data analysis. 

The amount of the mAb protein was calculated using the light-to-heavy peptide ratio. Data are presented as mean ± SEM. Data were compared using two-way ANOVA with Sidak’s correction. Statistical significance of the results from biodistribution studies was set at *p* < 0.05. Statistical analysis was performed using GraphPad 9.3.1 (GraphPad Software, Inc., Boston, MA, USA).

## Data Availability

The data supporting the findings of this study are available within the article (and/or its Appendix A).

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
