# Peer review of "Evaluation of Novel B1R/B2R Agonists Containing TRIOZAN™ Nanoparticles for Targeted Brain Delivery of Antibodies in a Mouse Model of Alzheimer Disease"

_molecules, 2023, doi:10.3390/molecules28135206_

Round 1
Reviewer 1 Report
This paper describes the preparation of Np trimethyl chitosan containing mouse mAb to reduce the activation of complement by the classical pathway (C1q), plus two peptides synthetic agonists receptor kinin, in order to open temporally and site-specifically the cerebral vasculature and allow the entry of Np to the cerebral parenchyma. The routes of administration were IV and intranasal. The agonists were synthesized, attached to a spacer arm, separated from the solid phase and linked in different proportions to the TMC chain through an arm. Different formulations were prepared, with different proportions of BR1 where BR 2 agonist [NG291] was free, as was the antibody. The presence of mAb was then sought by LC MS/MS of fractions of the brain (and reminder tissues?).
The manuscript contains a large volume of complex information and proposes an elaborate strategy whose effectiveness is still unknown. It constitutes a valuable contribution to the field of preclinical research. Some issues that should be clarified are presented below:
1) The chitosan NP are not water soluble, because the nanoparticles do not dissolve in the medium forming a homogeneous phase. Since they maintain their nanoparticulate character, it is more correct to say they can be dispersed or suspended in water.
2) The BR2 agonists [NG291] would be buried in the TMC NP. If so, and it still works, why add spacer arms in for BR1 agonist? This is important since would make a simpler formulation.
3) Why use a radioassay for competition binding assays? A flow cytometry would have been less complex. Or did you consider the inclusion of a fluorescent tag problematic?
4) Section 3.3. Preparation of TRIOZAN-based NPs for mAb loading: the authors simply add a quantity of mAb and then quantify proteins by spectrophotometry. What % of mAb is associated with Np and how reproducible is the technique? And how do the authors know that the mAb did not leak out of the NPs after IV injections?
5) Section 3.5. Quantification of mAb: explain why the authors do not consider the contribution of peptide agonists to the absorbance; furthermore, the presence of NP does not interfere with scattering or reducing mAb absorbance.
6) Provide a reasonable explanation for the fact that there is virtually no hepatic uptake of Np. This contradicts all experimental data regarding NP biodistribution.
Author Response
1) The chitosan NP are not water soluble, because the nanoparticles do not dissolve in the medium forming a homogeneous phase. Since they maintain their nanoparticulate character, it is more correct to say they can be dispersed or suspended in water.
Reply. Chitosan is different than TMC, which is more ionic therefore more soluble in water (see introduction, line 76). Yes, the correct nomenclature is a suspension of nanoparticles to be distinguished from a solution of molecules or macromolecules. In this work, the nanoformulations prepared are made by precipitating a solution of modified chitosan (trimethyl chitosan = Triozan), which is considered soluble in water even in basic media, with an anionic polymer (dextran sulfate) and a gelation agent (TPP) by controlled mixing. The nanoprecipitates thus formed in the presence of the antibodies and peptides are in suspension in the aqueous media and the nanoparticles are stabilized through cationic charge repulsion. In the manuscript we make the difference between polymer solutions and nanoparticle (NP) suspensions. The modified chitosan (Triozan) is in solution in PBS as it has not yet been precipitated. See also modified text, line 198.
2) The BR2 agonists [NG291] would be buried in the TMC NP. If so, and it still works, why add spacer arms in for BR1 agonist? This is important since would make a simpler formulation.
Reply. The reviewer is right, but this information could not be discovered without the current work in which the absence or presence of spacer arms could be compared. The B1R agonist used in F6 (referred to as NG29) does not contain a spacer arm (see Materials & Methods, line 362). In the case of F4, the spacer arm containing Mpa is required for the coupling of the B1R agonist to TMC.
3) Why use a radioassay for competition binding assays? A flow cytometry would have been less complex. Or did you consider the inclusion of a fluorescent tag problematic?
Reply. The competition radioligand binding assay is relatively easy to perform. Indeed, one key advantage over FACS assays is that it does not require the construction of fluorescent derivatives (conjugation of fluorescent tag on peptides) of the tested nanoformulations.
4) Section 3.3. Preparation of TRIOZAN-based NPs for mAb loading: the authors simply add a quantity of mAb and then quantify proteins by spectrophotometry. What % of mAb is associated with Np and how reproducible is the technique? And how do the authors know that the mAb did not leak out of the NPs after IV injections?
Reply. The pH of the Triozan solution was adjusted to pH > 9 using bicarbonate to promote electrostatic association between the positively charged Triozan (highly cationic, > 50% of quaternary ammonium groups) and the anti-C1q antibody which has a calculated isoelectric point of pI = 6.1 (negatively charged at pH 9). As there is no covalent bond between the Triozan polymer and the antibodies, the antibodies are trapped in the nanoparticles by electrostatic interactions and through physical entrapment in the mesh formed by oppositely charged polymers after nanoprecipitation. In the stock solution before injection, there are two fractions of the antibody, the antibody within the nanoparticles and the antibody outside of the nanoparticles, both fractions are in equilibrium. Upon intravenous injection, due to dilution of the nanoformulations, the equilibrium is shifted towards the unassociated antibody, the entrapped antibodies will diffuse slowly out of the porous nanoparticles. The ratio of free/nanoparticle-encapsulated antibody and the kinetic of release has not been established as the antibody quantity was limited and the detection technique by absorbance requires a relatively high level for detection. However, the accumulation to the brain tissue is faster than the release kinetic, so that a significant portion of the nanoparticle-associated antibody crossed the BBB before being completely released. We can expect that the fraction of the mAb which was released from the nanoparticles after the IV injection would be biodistributed in the same organs as the free mAb control groups, which is significantly low in the brain compared to the nanoencapsulated mAb.
5) Section 3.5. Quantification of mAb: explain why the authors do not consider the contribution of peptide agonists to the absorbance; furthermore, the presence of NP does not interfere with scattering or reducing mAb absorbance.
Reply. The peptide has low quantities of aromatic amino acids (only Phenylalanine, no Tryptophan nor Tyrosine), which contribute most to the absorbance at 280 nm. Tryptophan is the amino acid which has the highest absorptivity at 280 nm (5600 L.mol-1.cm-1 at 280 nm) compared to Tyrosine (1400 L.mol-1.cm-1 at 274 nm) or Phenylalanine (200 L.mol-1.cm-1 at 254 nm), for this reason we can neglect the contribution of the peptides to the total absorbance compared to the antibody (28 Tryptophans and 52 Tyrosines/antibody).
The contribution from the scattering of empty nanoparticles measured at similar concentrations was taken into consideration before measuring the mAb concentration from the 280 nm absorbance.
6) Provide a reasonable explanation for the fact that there is virtually no hepatic uptake of Np. This contradicts all experimental data regarding NP biodistribution.
Reply. We do not see a contradiction there. While the processing of nanoparticles is highly dependent on the composition of the nanomaterial, it is generally true that untargeted (surface unmodified) nanoparticles will undergo greater accumulation and sequestration in the liver than targeted NPs after their IV administration. In our study, we used stabilized, potent kinin B1R/B2R agonists to promote their selective delivery to the brain, as emphasized in the text (see lines 247-250).
Reviewer 2 Report
The reviewed manuscript concerns new nanoformulations containing biobased Triozans polymers along with peptide kinin B1R and B2R agonist analogs, as potent BBB-permeabilizers, to enhance brain delivery potentially useful in the treatment of Alzheimer's disease. The manuscript is of interest to a wide range of scientists specializing in the design of new drugs and drug delivery systems, especially those that should be delivered to the brain.
In my opinion, the manuscript needs small corrections i.e. units should be added when presenting molar masses (in Table 1 and in the text).
Author Response
In my opinion, the manuscript needs small corrections i.e. units should be added when presenting molar masses (in Table 1 and in the text).
Reply. Modifications made as suggested.
Reviewer 3 Report
“Evaluation of novel B1R/B2R agonists containing TRIOZANTM nanoparticles for targeted brain delivery of antibodies in a mouse model of Alzheimer disease”.
Maxime Gagnon, et al performed this study by exploring the Evaluation of novel B1R/B2R agonists containing TRIOZANTM nanoparticles for targeted brain delivery of antibodies in a mouse model of Alzheimer disease. The manuscript is well designed and written in scientific way. However, there are many major concerns/flaws and areas which should be addressed before publishing it.
1. Abstract: Certain terms that appear 1st should be abbreviated like B1R, B2R, Tg-SwDI mouse model, etc to clear the idea for readers.
2. The abstract is presented somewhat in general having no prper explanations of the key results.
3. The author should amend in abstract some numerical values from the key results.
4. Introduction: The 1st para of introduction section is from line 36 to 42 is very much deficient of coding appropriate references. A detail background of Alzheimer’s disease should be supported by an appropriate reference as it did not specify the direction for the readers; for guidance; https://doi.org/10.3390/molecules27082468,
https://doi.org/10.3390/ph15101205
https://doi.org/10.1097/cm9.0000000000001706,
https://doi.org/10.1016/j.fitote.2022.105268.
5. Line 102; what is ANX005? It did not appear in abstract? Give its brief description in text.
6. Line 142; IC50 should be written as IC50 and should be uniform in all around the text.
7. The author claims in line 171, 172, 173 and 174 “ligands trapped inside the NPs are in no position to meet their receptors counterparts. Therefore, a loss of affinity is to be expected in that situation as indeed observed in nano formulation F4, where the B1R ligand only is covalently linked to the NP, whereas the B2R ligand is not. “The author should discuss this claim in the light updated literature.
8. In the results section most parts were did not discuss properly, that should be properly discuss supporting relevant literature.
9. In my opinion it would be better if the author separate the discussion section from the results.
10. Line 328; Mw=65,000 should be written as Mw = 65,000 and should be uniform throughout the text.
11. Line 314, 335 and 383; -20oC should be written as -20 oC with space and keep it uniform.
12. Methods; section 3.3; this method is performed according to the previous reported protocol which should be reflected via reference.
13. Support section 3.6, 3.8 and 3.9 with appropriate references
14. Line 424; -80oC should be -80 oC.
15. Conclusion of the study is missing, it is important to draw conclusion of the study in the manuscript.
16. What breakthroughs do the authors think their research has made compared with the past?
17. There are many flaws found in the manuscript which should be carefully addressed before publishing.
18. In my opinion this is a good study having pronounced future perspectives; however, many points as well as typos error should be addressed before publishing it.
Extensive editing of English language required
Author Response
1) Abstract: Certain terms that appear 1st should be abbreviated like B1R, B2R, Tg-SwDI mouse model, etc to clear the idea for readers.
Reply. Some abbreviations have now been defined. However, by doing so, the word count limit of 200 established by the journal, has been exceeded (new word count: 205 words)
2) The abstract is presented somewhat in general having no proper explanations of the key results.
Reply. See reply to comment #1.
3) The author should amend in abstract some numerical values from the key results.
Reply. See reply to comment #1.
4) Introduction: The 1st para of introduction section is from line 36 to 42 is very much deficient of coding appropriate references. A detail background of Alzheimer’s disease should be supported by an appropriate reference as it did not specify the direction for the readers.
Reply. We have included a new reference to support general information related to AD (see line 41).
5) Line 102; what is ANX005? It did not appear in abstract? Give its brief description in text.
Reply. The ANX005 is now mentioned in the abstract. For the description of ANX005, see introduction, lines 62-63.
6) Line 142; IC50 should be written as IC50 and should be uniform in all around the text.
Reply. Correction made as suggested.
7) The author claims in line 171, 172, 173 and 174 “ligands trapped inside the NPs are in no position to meet their receptors counterparts. Therefore, a loss of affinity is to be expected in that situation as indeed observed in nano formulation F4, where the B1R ligand only is covalently linked to the NP, whereas the B2R ligand is not. “The author should discuss this claim in the light updated literature.
Reply. Since we did not compare the binding capacity of F4 and F6 at B1R and B2R, we removed that claim from the text (see lines 172-175).
8) In the results section most parts were did not discuss properly, that should be properly discuss supporting relevant literature.
Reply. We believe that the results/discussion section, which contains 15 references, is adequately presented, and supported by sufficient references.
9) In my opinion it would be better if the author separate the discussion section from the results.
Reply. This format is allowed by the journal. It has also been accepted by the other two reviewers.
10) Line 328; Mw=65,000 should be written as Mw = 65,000 and should be uniform throughout the text.
Reply. Correction made as suggested.
11) Line 314, 335 and 383; -20oC should be written as -20 oC with space and keep it uniform.
Reply. Corrections made as suggested.
12) Methods; section 3.3; this method is performed according to the previous reported protocol which should be reflected via reference.
Reply. Protocols for NPs fabrication were developed by our collaborators (A.M. and P.T.) at Ovensa Inc. The method was not reported before.
13) Support section 3.6, 3.8 and 3.9 with appropriate references
Reply. Protocols for DLS analysis (section 4.6) were under recommended instrument manufacturer’s instructions. Protocols for the biodistribution studies (section 4.8) and mAb administration (section 3.9) were conjointly developed by our laboratory and that of Ovensa Inc.
14) Line 424; -80oC should be -80 oC.
Reply. Correction made as suggested.
15) Conclusion of the study is missing. It is important to draw conclusion of the study in the manuscript.
Reply. Conclusions are not missing and can be found just after the Materials & Methods section, according to the manuscript formatting guidelines. For the convenience of the readers, the text of the conclusion was moved at the end of the Results/Discussion section (see revised version). We hope that this modification will be found acceptable by the Editor.
16) What breakthroughs do the authors think their research has made compared with the past?
Reply. See responses to comment #15.
Round 2
Reviewer 3 Report
The suggestions incorporated.
Minor editing of English language required